# Beyond GPS: Tri-Modal Contrastive Learning for Global Geo-Localization

## Abstract

Global image geo-localization aims to predict the precise geographic location of a photo anywhere on Earth based on a single image. This task is highly challenging yet widely applicable, especially in GPS-denied scenarios such as robotic navigation, post-disaster rescue, and open-world understanding. Existing methods often overlook the geographic information embedded in the language modality, making it difficult to resolve visual ambiguity and handle the heterogeneous global image distribution. To address these issues, we propose a unified image–text–GPS tri-modal contrastive learning framework to enhance the robustness and accuracy of global geo-localization. We first construct a high-quality tri-modal annotation pipeline that integrates semantic segmentation, visual-language generation, and a referee mechanism to automatically generate image-level and region-level descriptions. Geographic labels such as city and country names are also introduced as textual supplements. We then design a unified projection space where image, text, and GPS coordinates are embedded into a shared representation. A dual-level contrastive learning strategy at both global and regional scales is employed to strengthen semantic–spatial alignment across modalities. In addition, we introduce a hierarchical consistency loss and a dynamic hard negative mining strategy to further enhance representational discrimination and structural stability. Experimental results demonstrate that our method surpasses existing state-of-the-art approaches on multiple public geo-localization benchmarks, including Im2GPS3k, GWS15k, and YFCC26k, validating the effectiveness and generality of tri-modal alignment for global image geo-localization.

## 1 Introduction

Worldwide Image Geo-localization (WIGL) Vo et al. (2017) aims to precisely predict the shooting location of any photo taken on Earth. Unlike local geolocalization (e.g., at the city level) Tan et al. (2021), geolocalization at the global scale greatly expands the application potential of this task, supporting a wide range of real-world scenarios such as navigation, tourism, security, and crime tracking. However, the task remains highly challenging: images collected worldwide exhibit substantial diversity (e.g., landscapes, weather conditions, architectural styles), and localization becomes particularly difficult in regions lacking landmarks or outside popular areas.

Global image geolocalization methods on a global scale can be broadly categorized into three types: (1) Classification-based methods Clark et al. (2023); Pramanick et al. (2022): partition the Earth into discrete geographical units and train classifiers using hierarchical structures, scene-level tags, or regional parsing; (2) Retrieval-based methods Yang et al. (2021); Zhu et al. (2022; 2023): localize images based on the known coordinates of visually similar images in a reference gallery, but scaling to the global level requires maintaining an impractically large gallery; (3) Retrieval-augmented generation (RAG) methods Zhou et al. (2024); Jia et al. (2024): leverage the reasoning ability of large multimodal models (LMMs) by injecting retrieved GPS coordinates into prompt templates to generate more accurate predictions. Global image geolocalization still faces key challenges. First, geographically distant regions may share similar visual appearances, making it difficult for traditional visual features to capture fine-grained spatial semantics. Second, geographic imbalance in global datasets leads retrieval-based methods to perform unstably in data-sparse or remote regions. Moreover, existing approaches generally overlook the semantic cues contained in image descriptions. We argue that integrating language information with visual features through multimodal modeling is crucial for improving the accuracy and robustness of global image geolocalization.

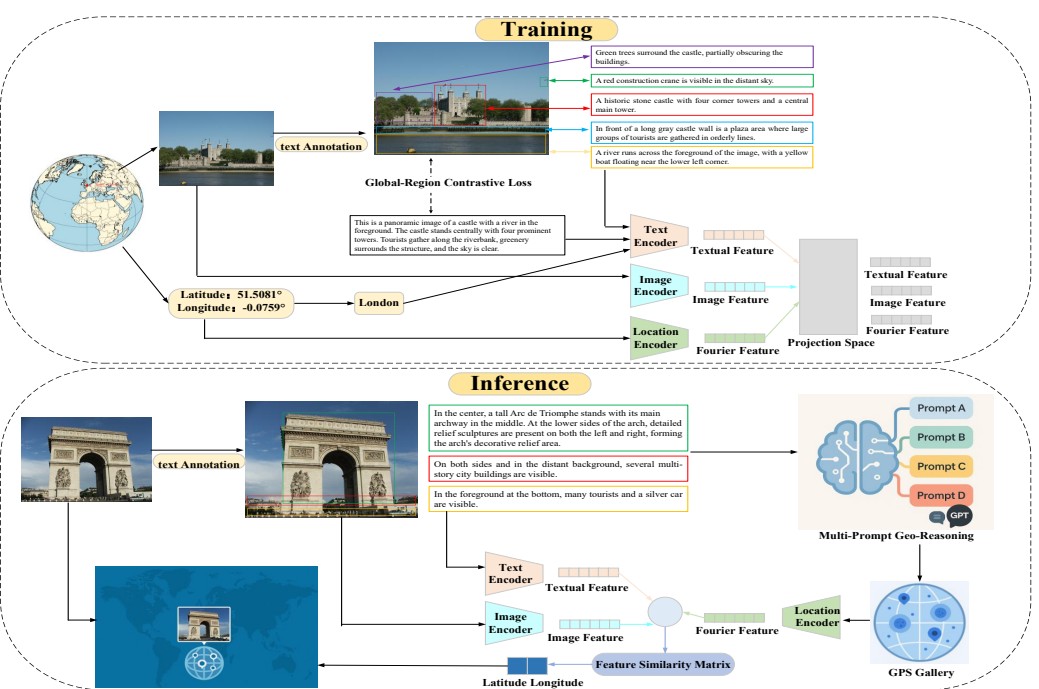

Figure 1: Overview of the Framework. First, the input image passes through an automated annotation pipeline to produce global and regional semantic descriptions, while its geographic coordinates are reverse-geocoded into country, region, and city names. During training, GPS coordinates are mapped to high-dimensional Fourier features, image and text features are extracted with CLIP encoders, and all three are projected into a shared embedding space and jointly optimized with global and local contrastive losses. At test time, we introduce Multi-Prompt Reasoning, using diverse prompts to generate multiple candidate locations. For each candidate, we sample surrounding GPS points to expand a GPS Gallery. Finally, we match the query's image/text features against the gallery embeddings and select the coordinate with the highest similarity.

We propose an automated annotation pipeline to construct high-quality image–text–GPS tri-modal training data. Specifically, a semantic segmentation model extracts meaningful regions, a vision–language model (VLM) generates global and local descriptions, and—given VLM hallucinations—we introduce a Referee Model that filters low-quality or hallucinated content via keyword verification and regeneration, producing structured textual annotations and geographic labels for each image. Building on this, we present the Geo-alignment framework for tri-modal retrieval and alignment. The framework maps GPS coordinates into high-dimensional embeddings and projects them, together with image and text features, into a unified embedding space to enable cross-modal representation learning. Finally, we design a multi-prompt–driven geographic reasoning module. For each query image, we feed a set of prompts to a large language model to generate diverse candidate locations and dynamically expand the GPS gallery. The final prediction is obtained by measuring the similarity between the query's visual/text features and the geographic embeddings of all candidate coordinates and selecting the best match.

Our main contributions are as follows: 1. We propose a multi-modal contrastive learning framework based on image–text–GPS data for worldwide image geolocalization. 2. We construct a novel image–text–GPS benchmark dataset, which augments existing image–GPS datasets with geographic labels as well as image-level and region-level textual descriptions. 3. We design an image annotation framework that integrates semantic segmentation, vision–language generation, and a Referee model filtering mechanism. 4. We propose a multi-prompt-driven geographic reasoning module that guides LLMs to generate multiple candidate locations and dynamically expand the GPS coordinate gallery. The overall framework is illustrated in the Fig. 1.

## 2 RELATED WORK

### 2.1 LOCATION PREDICTION FROM IMAGES

In image geolocation, early methods relied on image retrieval Muller-Budack et al. (2018); Pramanick et al. (2022), but their dependence on large landmark databases makes global deployment

infeasible. Classification-based approaches Clark et al. (2023); Pramanick et al. (2022) partition the earth into geo-cells and refine them with hierarchical categories, multi-level modeling or semantic fusion. However, fixed class centers and discrete boundaries limit prediction accuracy in real-world scenarios. To address this, we formulate geolocation as direct regression from images to GPS coordinates using continuous spatial representations. We apply equal-earth projection Lapaine & Frančula (2022) to reduce high-latitude distortion and encode GPS with multi-scale learnable Fourier features Li et al. (2021), mitigating the spectral bias of MLPs and enabling finer localization.

## 2.2 MULTIMODAL GENERATION AND LLM-ASSISTED ANNOTATION

In recent years, large language models (LLMs) have demonstrated strong capabilities in multimodal representation learning Li et al. (2024); Zhang et al. (2024); Wu et al. (2023). Flamingo Alayrac et al. (2022) and BLIP-2 Li et al. (2023) achieve image–text alignment and open-domain question answering via cross-modal attention; general LLMs such as GPT-4 are used for prompted image captioning Zhang et al. (2023); Maniparambil et al. (2023), effectively reducing manual annotation costs. Meanwhile, contrastive learning methods like CLIP Hafner et al. (2021); Zhang et al. (2022) construct a unified image–text semantic space. However, most existing studies focus on the image–text modality, with limited exploration of jointly modeling LLMs and geographic coordinates. Notably, the Perceive Anything Model Lin et al. (2025) unifies region segmentation and semantic generation, showcasing the synergistic potential of LLMs and region-level visual understanding. Its "region perception + semantic generation" paradigm provides important inspiration for geographic semantic modeling. We adapt PAM's idea to the global localization task by annotating different regions in geographic image datasets. As the foundation of our vision–language annotation pipeline, the generated geographic tags are jointly modeled with image and coordinate embeddings to construct a unified tri-modal space. This method realizes the fusion prediction of image, text and location modalities in global geolocation tasks for the first time, demonstrating the wide application potential of generative language models in spatial understanding tasks.

## 3 DATASET ANNOTATION PIPELINE

To construct high-quality supervisory signals for tri-modal alignment across images, text, and geographic coordinates, we design a vision-language annotation pipeline. This pipeline operates at two levels of granularity—global (image-level) and local (region-level)—and integrates semantic segmentation, vision-language model (VLM) generation, automatic evaluator verification, and keyword-based filtering. Through multi-stage validation and refinement, this pipeline effectively mitigates hallucinations and semantic inconsistencies, thereby ensuring consistency and reliability among the annotated text, visual content, and geographic coordinates.

### 3.1 DATASET ANNOTATION DESCRIPTION

Our proposed annotation method extends the Im2GPS3k Hays & Efros (2008), GWS15k Clark et al. (2023), YFCC26k Thomee et al. (2016) dataset by introducing fine-grained vision-language-geography annotations. Specifically, region-level annotations: Each image contains an average of 4.5 bounding boxes, with each box paired with a region-level natural language description averaging 15.2 words, providing precise correspondence to local visual content. **Global descriptions:** Each image also includes a comprehensive description covering both global and local details, averaging 61.4 words. **Geographic labels:** Beyond visual-language annotations, we further incorporate discrete geographic coordinate labels, enabling tri-modal alignment between images, language, and geospatial information. As illustrated in Fig. 5, compared with the original dataset, our proposed dataset not only incorporates more fine-grained region-level annotations but also integrates natural language and geographic information. This enriched design is crucial for natural language-guided geo-localization and navigation tasks.

### 3.2 IMAGE AND REGION-LEVEL ANNOTATION

As shown in Fig. 4, the pipeline consists of three stages:

**Semantic Region Extraction:** Given a raw image collection $\mathcal{I} = \{I_i\}_{i=1}^{M}$, we apply a pretrained semantic segmentation model (e.g., SAM + Mask2Former) to extract pixel-level masks: $R_i = \{r_{i,j}\}_{j=1}^{N_i}$, s.t. $r_{i,j} \subset I_i$, where $r_{i,j}$ denotes the $j$-th valid semantic region within image $I_i$. We further filter low-confidence or extremely small regions based on mask area and prediction scores to improve region description quality.

**Initial Vision-Language Generation:** We employ a vision-language model (VLM) to generate initial textual descriptions for each image and its regions: 1. Global-level description: For each whole image $I_i$, we generate $K_g$ textual candidates: $\mathcal{T}_i^{\text{global}} = \{\text{VLM}(I_i, p_g^{(k)})\}_{k=1}^{K_g}$, where $p_g^{(k)}$ denotes the

$k$-th global-level prompt (e.g., "Please describe this scene concisely"). 2. Region-level description: For each segmented region $r_{i,j}$, we generate a region-level caption: $\mathcal{T}_{i,j}^{\text{region}} = \text{VLM}(r_{i,j}, p_r)$ where $p_r$ prompts the model to focus on the target region and include spatial references (e.g., "Describe this region and its position relative to the image center").

**Referee Model for Automatic Validation:** Large vision-language models still exhibit common issues such as hallucination, vague phrasing, and semantic misalignment. As such, raw textual outputs are not guaranteed to meet annotation standards. To address this, we introduce a *Referee Model* to automatically validate the quality of VLM-generated descriptions. The Referee model applies a two-stage filtering process: **Positive keyword detection:** Validates whether the candidate text includes at least one predefined semantic keyword, ensuring spatial or entity information is present. **Negative sample exclusion:** Scans for undesirable tokens using a blacklist of negative keywords, which capture hallucinated or ill-formed content.

The construction of keyword sets is guided through human-in-the-loop curation. Specifically, we sample approximately 5,000 initial VLM outputs and feed them into a secondary large language model (serving as a "teacher model") for binary classification. Based on the teacher's outputs, we build frequency-based keyword lists: **Negative keyword list** $\mathcal{K}^-$: includes common failure patterns such as HTML tags (e.g., "img src", "[image]"), apology phrases (e.g., "sorry"), and URLs. **Positive keyword list** $\mathcal{K}^+$: contains spatial or positional expressions (e.g., "on the left", "next to the building", "far from the road") to ensure geographic relevance. This referee mechanism enables automatic screening and regeneration of VLM outputs, significantly reducing the need for manual intervention while enhancing the structural consistency and semantic utility of the final annotations.

### 3.3 Reverse Geocoding

On the other hand, discrete geographic labels such as continent, country, city also play a crucial role in geo-localization. Compared to continuous GPS coordinates, these semantically explicit discrete attributes often exhibit abrupt transitions at geographic boundaries, making them an effective supplement to the spatial localization of images. To convert latitude and longitude into location names, we introduce a reverse geocoding module based on public geographic databases (e.g., Nominatim or Google Geocoding API), which parses each image's GPS coordinates $(\text{lat}_i, \text{lon}_i)$ to obtain the corresponding country, region, and city names, denoted as: $\text{Loc}_i = [\text{Continent}_i, \text{Country}_i, \text{City}_i]$. We then transform the location tags into human-readable natural language text as follows: $T_i^{\text{geo}} =$ "A photo taken in " $+ \text{Continent}_i + \text{Country}_i + \text{City}_i$, The geographic description $T_i^{\text{geo}}$ is concatenated with the content-based image description $T_i^{\text{desc}}$ to form the complete textual input: $T_i^{\text{full}} = T_i^{\text{desc}} + T_i^{\text{geo}}$. We transform numerical GPS coordinates into text expressions with regional semantics, making them more easily aligned with visual content during multi-modal learning.

## 4 Multimodal contrastive learning

### 4.1 Image and Text Encoders

In this work, we adopt the CLIP Radford et al. (2021) model as the backbone encoder for both image and text modalities, leveraging its general semantic representations learned from large-scale image-text alignment tasks. For the image encoder, we employ the ViT-L/14 from CLIP as the backbone and keep it frozen to preserve its original visual understanding capabilities. Similarly, for the text encoder, we use the text encoder from CLIP and also keep its parameters frozen.

### 4.2 Location Encoder

In worldwide image geo-localization, GPS coordinates $\mathbf{G} = (\text{lat}, \text{lon}) \in \mathbb{R}^2$ are low-dimensional continuous signals, and feeding them directly into an MLP makes alignment with image or text embeddings difficult. Their dimensionality is far lower than the high-dimensional features produced by models like CLIP; moreover, low-dimensional inputs are susceptible to spectral bias Tancik et al. (2020); Vivanco Cepeda et al. (2023), tending to learn only low-frequency patterns. More critically, in high-dimensional spaces used by contrastive learning (e.g., InfoNCE, triplet loss), low-dimensional coordinate features both struggle to align and can disrupt optimization due to scale mismatch, leading to unstable training. To address this, we map GPS coordinates to high-dimensional, structured representations. The position encoder comprises three components: Equal-Earth projection, multi-scale learnable Fourier features, and unified feedforward fusion.

### 4.2.1 Equal Earth Projection:

Conventional latitude-longitude coordinate systems suffer from nonlinear distortion in high-latitude regions, where Euclidean distances no longer correspond proportionally to real-world geographic distances. This mismatch can mislead the contrastive learning objective. To alleviate this, we apply

the Equal-Earth (EE) projection $\varphi(\cdot)$ to transform the spherical GPS coordinate $\mathbf{G} = (\text{lat}, \text{lon})$ into a 2D planar coordinate $\mathbf{z} = [x, y]^\top$ as follows:

$$G_i'^{\,\text{lon}} \;=\; P_4\,\theta^9 \;+\; P_3\,\theta^7 \;+\; P_2\,\theta^3 \;+\; P_1\,\theta \tag{1}$$

$$G_i'^{\,\text{lat}} \left[ \begin{array}{c} \dfrac{2\sqrt{3}\,G_i'^{\,\text{lon}}\cos\theta}{3\left(9p_4\theta^8 + 7p_3\theta^6 + 3p_2\theta^2 + p_1\right)} \\ p_4\theta^9 + p_3\theta^7 + p_2\theta^3 + p_1\theta \end{array} \right], \theta = \arcsin\left(\tfrac{\sqrt{3}}{2}\sin G_i'^{\,\text{lat}}\right), \tag{2}$$

where, the constants are defined as $p_1 = 1.340264$, $p_2 = -0.081106$, $p_3 = 0.000893$, and $p_4 = 0.003796$ according to the EE projection. After projection, the output coordinates $(x, y)$ are linearly normalized to the range $[-1, 1]$. This transformation ensures that Euclidean distances in the embedding space better reflect actual geographic distances, reducing the burden on the learning process to compensate for spatial distortions later.

### 4.2.2 Multi-Scale Learnable Fourier Features

Although the Equal-Earth projection maps geographic coordinates to a 2D planar space, the resulting vector $\mathbf{z} = [x, y]^\top$ remains low-dimensional and smooth. Directly feeding it into an MLP often leads to spectral bias, where the model tends to learn low-frequency structures and fails to capture fine-grained spatial differences. To address this, we adopt a multi-scale learnable Fourier encoding mechanism inspired by Random Fourier Features (RFF), which explicitly injects high-frequency signals into the coordinate representation. Our design incorporates three components: log-scale frequency sampling, orthogonal random bases, and learnable gated-phase Fourier mapping.

**Frequency Sampling.** To ensure coverage of spatial patterns from continental to street-level granularity, we uniformly sample $M$ scales in log space. The standard deviation $\sigma_i$ of each scale as: $sigma_i = \sigma_{\min}\left(/\sigma_{\max}\sigma_{\min}\right)^{\frac{i-1}{M-1}}$, $i = 1, \ldots, M$. This logarithmic sampling guarantees that the model captures spatial variations across multiple resolutions, mitigating the dominance of low-frequency components. **Orthogonal Random Bases.** For each frequency scale, we generate an orthogonal random matrix $R_i \in \mathbb{R}^{K \times 2}$ and apply frequency normalization to obtain the projection matrix: $B_i = \sigma_i^{-1} R_i$ where $K$ is the number of frequency channels per scale. Orthogonality reduces feature redundancy and enhances frequency diversity.

**Gated Phase Fourier Mapping:** Given the projected coordinate $\mathbf{z}$, the Fourier features at scale $i$:

$$\Gamma_i(\mathbf{z}) = \begin{bmatrix} \mathbf{g}_i \odot \sin(2\pi B_i \mathbf{z} + \boldsymbol{\delta}_i) \\ \mathbf{g}_i \odot \cos(2\pi B_i \mathbf{z} + \boldsymbol{\delta}_i) \end{bmatrix} \in \mathbb{R}^{2K} \tag{3}$$

$g_i = \text{softplus}(w_i)$ is a learnable gating weight for each channel, where the softplus activation ensures non-negativity. $\boldsymbol{\delta}_i \in \mathbb{R}^K$ is the learnable phase shift. The symbol $\odot$ denotes element-wise multiplication. This encoding scheme not only explicitly introduces high-frequency components but also enables the network to automatically select the most informative frequency dimensions and adjust their phases, thereby enhancing representational flexibility and data adaptability. The final Fourier feature is formed by concatenating features across all scales: $\Gamma(\mathbf{z}) = [\Gamma_1(\mathbf{z}), \ldots, \Gamma_M(\mathbf{z})]$. Multi-scale modeling mitigates spectral bias, enabling the network to capture both global and local spatial variations. The gating mechanism ($\mathbf{g}_i$) softly selects among frequency components to suppress noisy dimensions and improve robustness, while the learnable phase shift ($\boldsymbol{\delta}_i$) introduces translation equivariance and alleviates gradient vanishing near sinusoidal zero-crossings. This encoder provides a high-dimensional structured representation of GPS coordinates, serving as a key component of our tri-modal contrastive learning framework.

### 4.2.3 Unified Feedforward Aggregation

Building upon the multi-scale encoding, prior works design separate MLP branches for each frequency band. However, such multi-branch structures often lead to computational redundancy and gradient inconsistency, increasing model complexity and impairing training stability. To address these issues, we propose a unified feedforward network to aggregate the frequency-domain features from all scales and produce the final coordinate embedding. Formally, given an input coordinate $\mathbf{G}$, we apply the Equal-Earth projection followed by multi-scale Fourier encoding to obtain $\Gamma(\varphi(\mathbf{G})) \in \mathbb{R}^{2KM}$. We then employ a two-layer feedforward neural network defined as: $\mathbf{L}(\mathbf{G}) = \mathbf{W}_2 \cdot \sigma\left(\text{LN}(\mathbf{W}_1 \cdot \Gamma(\varphi(\mathbf{G})))\right) \in \mathbb{R}^d$, where $\mathbf{W}_1 \in \mathbb{R}^{h \times 2KM}$ and $\mathbf{W}_2 \in \mathbb{R}^{d \times h}$ are trainable weight matrices, $\sigma(\cdot)$ denotes the GELU activation function, and LN is Layer Normalization. This design maintains structural simplicity while providing sufficient expressive power. The unified feedforward network adaptively learns nonlinear combinations of frequency features, and the

single-path architecture effectively mitigates gradient fragmentation in multi-branch networks, improving convergence stability. Ultimately, this module produces consistent coordinate embeddings that support cross-modal contrastive learning.

## 4.3 Unified Projection Space Construction

We extract image and text features using CLIP, so their feature modalities are naturally well-aligned. For GPS coordinates, a positional encoder is employed to lift the raw 2D input into a high-dimensional representation. However, the resulting coordinate embeddings differ fundamentally from the CLIP space in both semantic structure and distribution. Without additional alignment, directly applying contrastive objectives leads to biased similarity computation and unbalanced gradient propagation, thereby undermining the training process of tri-modal contrastive learning. To address this issue, we introduce a unified projection space that maps all three modalities into the same latent space. $\hat{\mathbf{v}}_I \in \mathbb{R}^{d_I}$, $\hat{\mathbf{v}}_T \in \mathbb{R}^{d_T}$, $\hat{\mathbf{v}}_G \in \mathbb{R}^{d_G}$, where $d_I \gg d_G = 2$ denotes the dimensionality of image, text, and GPS features respectively. We define a shared semantic space $\mathcal{Z} \subset \mathbb{R}^d$ with $d = 512$, and construct a two-layer feed-forward network for each modality:

$$P_m(\hat{\mathbf{v}}_m) = \mathbf{W}_{m,2} \cdot \sigma\left(\text{LN}(\mathbf{W}_{m,1} \cdot \hat{\mathbf{v}}_m)\right), \quad m \in \{I, T, G\}, \tag{4}$$

where $\mathbf{W}_{m,1} \in \mathbb{R}^{h \times d_m}$ and $\mathbf{W}_{m,2} \in \mathbb{R}^{d \times h}$ are learnable weights, $\sigma(\cdot)$ is the GELU activation function, and $\text{LN}(\cdot)$ denotes Layer Normalization. To maintain the original CLIP image-text alignment and maximize parameter sharing, we set the image and text branches to share projection weights: $\mathbf{W}_{I,i} = \mathbf{W}_{T,i}$, $i = 1, 2$, while the coordinate branch remains independently parameterized to reduce training complexity. To alleviate potential gradient imbalance caused by inter-modality distributional gaps, we introduce a learnable modality-specific temperature vector $\boldsymbol{\tau} = (\tau_I, \tau_T, \tau_G)$ and apply a centralization-normalization scheme:

$$\mathbf{z}_m = \frac{P_m(\hat{\mathbf{v}}_m) - \boldsymbol{\mu}}{\|P_m(\hat{\mathbf{v}}_m) - \boldsymbol{\mu}\|_2} \cdot \tau_m, \boldsymbol{\mu} = \frac{1}{3}\sum_m \mathbb{E}[P_m(\hat{\mathbf{v}}_m)], \tag{5}$$

where $\boldsymbol{\mu}$ denotes the cross-modal feature mean and each temperature $\tau_m \in \mathbb{R}_+$ is jointly optimized via backpropagation. The resulting embeddings $\mathbf{z}_I, \mathbf{z}_T, \mathbf{z}_G \in \mathcal{Z}$ are thus aligned in dimensionality, scale, and norm, making them directly comparable for downstream tri-modal contrastive objectives. This projection framework not only preserves the semantic structure of the CLIP space but also explicitly guides GPS features to align with these semantics, ensuring a numerically stable and semantically meaningful representation for multimodal contrastive learning.

## 4.4 Global–Region Joint Tri-Modal Contrastive Learning

Existing multimodal contrastive learning mainly aligns image–text embeddings at the whole-image level, while real-world geographic scenarios require both global semantics and local structures. Relying solely on global features can confuse visually similar but geographically different regions, whereas focusing only on local features risks losing contextual information. Therefore, we propose a joint modeling approach with two granularity levels: (1) Global contrast provides robust scene-level semantic anchors to ensure large-scale cross-modal consistency; (2) Region-level contrast enhances contrast difficulty and fine-grained discriminability through hard positive and negative mining (intra-image inter-region, cross-image nearby regions). The hierarchical contrastive learning design not only alleviates the limitations of single-granularity learning but also enables a two-stage strategy of "coarse localization + fine adjustment," achieving higher resolution for local positioning while maintaining macro-level stability. We introduce a global–region joint tri-modal contrastive learning framework, which simultaneously optimizes alignment at the image level and region level in a shared projection space $\mathcal{Z} \subset \mathbb{R}^d$. For each input image $I_i \in \mathcal{I}$, we utilize a pretrained semantic segmentation model to obtain pixel-level masks and divide it into $R_i = \{r_{i,1}, r_{i,2}, \ldots, r_{i,N_i}\}$ semantic regions. Each region $r_{i,j}$ corresponds to an image region feature $\mathbf{v}_I^{(i,j)}$, a textual description feature $\mathbf{v}_T^{(i,j)}$, and its corresponding GPS embedding $\mathbf{v}_G^{(i,j)}$, which are projected to the shared embedding space via the projection module described in Section 3:

$$\mathbf{z}_m^{(i,j)} = \frac{P_m(\hat{\mathbf{v}}_m^{(i,j)}) - \boldsymbol{\mu}}{\|P_m(\hat{\mathbf{v}}_m^{(i,j)}) - \boldsymbol{\mu}\|_2} \cdot \tau_m, \quad m \in \{I, T, G\}, \tag{5}$$

$\tau_m \in \mathbb{R}_+$ denotes the learnable temperature parameter for modality $m$. To align semantic embeddings across three modalities, we introduce the following three loss terms:

**1. Global Contrastive Loss:** This term ensures semantic consistency among the entire image, the overall textual description, and the GPS coordinates by using the InfoNCE loss:

$$\mathcal{L}_{\text{global}}^{(i)} = - \sum_{(m,n)\in\mathcal{P}} \log \frac{\exp(\text{sim}(\mathbf{z}_m^{(i)}, \mathbf{z}_n^{(i)})/\tau)}{\sum_{k\neq i}\exp(\text{sim}(\mathbf{z}_m^{(i)}, \mathbf{z}_n^{(k)})/\tau)}, \tag{6}$$

where, $\mathcal{P} = \{(I,T),(I,G),(T,G)\}$, $\mathbf{z}_m^{(i)} \in \mathbb{R}^d$ denotes the normalized embedding vector of sample $i$ in modality $m \in \{I, T, G\}$, $\text{sim}(\cdot,\cdot)$ represents cosine similarity, and $\tau$ is the temperature coefficient. The set $\mathcal{P}$ enumerates all three modality pairs: image–text, image–GPS, and text–GPS. This loss encourages embeddings from the same image across different modalities to be close while pushing away negative samples from other images.

**2. Region Triplet Loss.** For each region $r_{i,j}$, in order to bring the tri-modal embeddings of the same region within the same image closer and push apart embeddings from different regions or images, we introduce a cross-modal triplet loss:

$$\mathcal{L}_{\text{region}}^{(i,j)} = \sum_{(m,n)\in\mathcal{P}} \Big[\alpha + d\big(\mathbf{z}_m^{(i,j)}, \mathbf{z}_n^{(i,j)}\big) - \min_{(i',j')\neq(i,j)} d\big(\mathbf{z}_m^{(i,j)}, \mathbf{z}_n^{(i',j')}\big)\Big]_+ \tag{7}$$

where, $\mathbf{z}_m^{(i,j)}$ denotes the embedding representation of the $j$-th semantic region in the $i$-th image under modality $m$, $[\cdot]_+$ indicates the hinge function, $\alpha$ is a margin hyperparameter, and $d(\cdot, \cdot)$ denotes the distance function (e.g., Euclidean or $1 - \text{cosine}$ distance). This loss encourages intra-region cross-modal embeddings to be close and distant from the embeddings of other regions or images.

**3. Hierarchical Alignment Loss.** To align the semantic representations of region-level and image-level embeddings, we define the following regularization term:

$$\mathcal{L}_{\text{hier}}^{(i)} = \sum_{m\in\{I,T\}} \left\| \mathbf{z}_m^{(i)} - \frac{1}{N_i}\sum_{j=1}^{N_i} \mathbf{z}_m^{(i,j)} \right\|_2^2 \tag{8}$$

where $\mathbf{z}_m^{(i)} \in \mathbb{R}^d$ represents the global image or text embedding vector of sample $i$ under modality $m \in \{I, T\}$, and $\mathbf{z}_m^{(i,j)}$ denotes the local embedding of the $j$-th semantic region. $N_i$ is the total number of semantic segmentation regions for sample $i$. This loss minimizes the Euclidean distance between the global embedding and the mean of its local region embeddings, encouraging local features to naturally aggregate toward the global center in the projected space, thus enhancing consistency and structural expressiveness of semantic representations. Overall Objective Loss:

$$\mathcal{L} = \frac{1}{M}\sum_{i=1}^{M} \left( \lambda_g \mathcal{L}_{\text{global}}^{(i)} + \lambda_r \sum_{j=1}^{N_i} \mathcal{L}_{\text{region}}^{(i,j)} + \lambda_h \mathcal{L}_{\text{hier}}^{(i)} \right) \tag{9}$$

where, $\lambda_g$, $\lambda_r$, and $\lambda_h$ are hyperparameters that control the weights of different loss components.

**Positive and Negative Sampling Strategy:** To enhance the discriminative power and semantic resolution of region-level contrastive learning, we design a structured positive-negative sampling strategy that leverages both local semantic structure and spatial priors to provide precise supervisory signals for trimodal alignment. Specifically, we define: **Positive Pairs:** For each semantic region $r_{i,j}$ in image $i$, we treat the projected trimodal embeddings $\mathbf{z}_I^{(i,j)}$, $\mathbf{z}_T^{(i,j)}$, and $\mathbf{z}_G^{(i,j)}$ as positive pairs. This design ensures alignment across modalities by: - Ensuring all three embeddings originate from the same image and the same region; - Maintaining semantic consistency among the modalities to form strong positive supervision. **Hard Negatives:** Real-world scenes often contain many locally similar distractors. Using randomly sampled negatives may make training too easy, leading to sparse gradients and overfitting. To address this, we introduce two types of hard negatives: - Intra-image hard negatives: These are embeddings from different regions $r_{i,j'}$ ($j' \neq j$) in the same image that are visually or geometrically similar to the target region. They test the model's ability to distinguish fine-grained semantic boundaries. - Cross-image hard negatives: These are selected from other images based on proximity in GPS space (e.g., neighboring blocks) or similar textual semantics. This helps prevent the model from relying solely on geographic closeness or superficial semantic cues. **Easy Negatives:** To maintain diversity and training stability, we also randomly sample unrelated regions across different images as easy negatives. These are visually and semantically dissimilar, helping to form clear contrastive boundaries in the early training stage.

**Hard Negative Memory Bank:** To dynamically maintain effective negative samples, we implement a memory bank that stores region embeddings from both current and previous batches. During each training iteration, we use nearest-neighbor search to retrieve the top-$K$ most similar embeddings to a given region as hard negatives: $\mathcal{N}^{\text{hard}}(i,j) = \text{Top-}K\left(\left\{d(\mathbf{z}_m^{(i,j)}, \mathbf{z}_n^{(i',j')}) \mid (i',j') \neq (i,j)\right\}\right)$.

where $d(\cdot, \cdot)$ denotes a distance metric. This mechanism enhances the adversarial nature and robustness of training, encouraging the model to focus on regions that are semantically ambiguous, spatially adjacent, or visually similar. By incorporating global and region-level trimodal contrastive learning with hard-aware sampling, our method preserves macro-level semantic alignment while enabling fine-grained structure recognition and geographic modeling. This provides a robust and discriminative feature foundation for trimodal geolocalization and retrieval.

### 4.5 MULTI-PROMPT GEO-REASONING

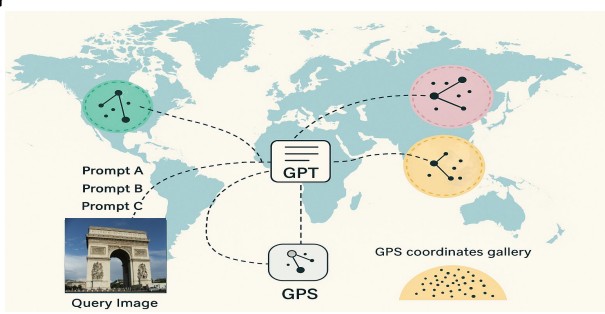

Figure 2: Overview of GPS Gallery Construction from Multi-Prompt Outputs.

To enhance the generalization and reasoning capability of large language models (LLMs) in geolocation tasks, we propose a prompt-driven framework named Multi-Prompt Geo-Reasoning (MPGR). MPGR aims to construct a semantically diverse and cognitively enriched set of geolocation candidates, enabling robust adaptation across a wide range of visual scenes. The overall framework is illustrated in Figure 3. MPGR consists of two main stages: Step 1: Image Candidate Mining (Positive/Negative Sampling) For each query image, we employ a vision-geolocation joint encoder (e.g., GeoCLIP) to retrieve a set of K geographically similar images from the training set as positive candidates. Additionally, we randomly select several geographically dissimilar images as negative candidates to strengthen the discriminative power of the subsequent reasoning process.

Step 2: Prompt Diversification for Geo-Prediction To fully exploit the multi-level reasoning capabilities of LLMs, we design a set of prompt templates with semantic diversity and contextual depth. These prompts guide the model to generate candidate geolocations from different inferential perspectives, forming a high-confidence pool of potential coordinates. Specifically, we define four representative types of prompts: **Prompt A:** Zero-shot Commonsense Reasoning leverages only the image content description and detected key objects to stimulate LLMs to perform open-ended geolocation reasoning, without relying on any external priors or exemplars. **Prompt B:** Comparative Inference provides the query image along with its retrieved positive candidates and associated descriptions. The model is guided to infer the likely location through analogy and semantic comparison. **Prompt C:** Contextual Landmark Decoding encourages the model to interpret multi-modal contextual clues, such as language, architectural styles, and terrain patterns, enabling better generalization across diverse regions. **Prompt D:** Fine-grained Urban Prediction focuses on micro-level urban cues (e.g., license plates, road signs, storefronts), guiding the model to reason at the street-block level for high-resolution localization. Through the coordinated guidance of these four types of prompts, MPGR constructs a rich and diversified set of candidate locations, providing reliable support for subsequent contrastive matching and fusion.

### 4.6 GPS GALLERY

We construct a hierarchically extended GPS coordinate database generation mechanism. In the initial stage, we build the base database through random sampling from the training set and image feature-based retrieval. Next, we leverage a set of designed prompts to guide the GPT model in generating multiple candidate geographic locations. For each

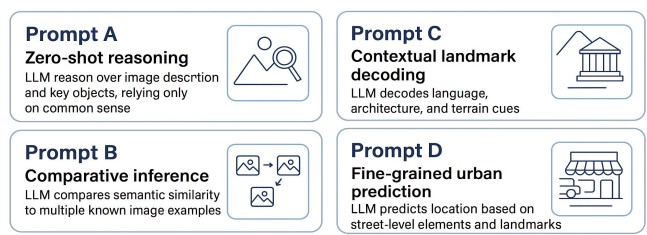

Figure 3: An overview of the four designed prompt strategies for guiding LLM in global geolocation tasks.

predicted location, we collect surrounding samples from the training set to further expand the GPS database, as shown in Fig. 2. To construct a globally distributed GPS coordinate database, we begin by sampling from the training set $\mathcal{D}_{\text{train}}$ with a priority on ensuring geographic uniformity. We divide the Earth's surface into uniform grid cells and randomly select samples from each grid cell, resulting in a base GPS set: $\mathcal{G}_{\text{base}} = \{(I_i, \text{GPS}_i) \mid (I_i, \text{GPS}_i) \in \mathcal{D}_{\text{train}}, \text{GPS}_i \in \text{UniformGrid}\}$. We incorporate candidate GPS generated by a series of prompt strategies $\{P_1, P_2, \ldots, P_K\}$. For each query image $I_q$, we apply each prompt $P_k$ to the LLM to obtain: $\mathcal{G}_q^{(k)} = \left\{ \hat{g}_{q,1}^{(k)}, \hat{g}_{q,2}^{(k)}, \ldots \right\}$. We then

retrieve training samples within a 25km radius of each predicted coordinate:

$$\mathcal{N}(\hat{g}_{q,j}^{(k)}) = \left\{ (I, \text{GPS}) \in \mathcal{D}_{\text{train}} \mid d(\text{GPS}, \hat{g}_{q,j}^{(k)}) \leq 25 \, \text{km} \right\} \tag{10}$$

where $d(\cdot, \cdot)$ is the geodesic distance. We aggregate all retrieved regions into a prompt-augmented pool: $\mathcal{G}_{\text{prompt}} = \bigcup_{q,k,j} \mathcal{N}(\hat{g}_{q,j}^{(k)})$, To further enhance diversity, we include visually retrieved samples $\mathcal{G}_{\text{retrieval}}$ based on image similarity (e.g., CLIP): $\mathcal{G}_{\text{final}} = \mathcal{G}_{\text{base}} \cup \mathcal{G}_{\text{prompt}} \cup \mathcal{G}_{\text{retrieval}}$. This multi-source construction results in a spatially rich and semantically diverse GPS database that facilitates robust and accurate geo-localization. **Inference:** To infer the geographic location of a query image, we first extract visual and textual features from the image and its corresponding textual description using a visual encoder and a text encoder. Meanwhile, each candidate location in the GPS coordinate database is transformed into a high-dimensional geographic representation via a GPS encoder. Then, in a shared embedding space, we compute the cosine similarity between the query's image-text features and each GPS embedding. Finally, a weighted aggregation is applied, and the GPS coordinate with the highest overall similarity is regarded as the most likely location of the query image.

## 5 Experiment

### 5.1 Datasets and Evaluation Details

For training, we use the MP-16 dataset Larson et al. (2017), which contains approximately 4.72 million geotagged images collected from Flickr. For testing, we evaluate the generalization ability of our model on several datasets, including Im2GPS3k Hays & Efros (2008), the recently introduced Google World Streets 15K (GWS15k) Clark et al. (2023), and YFCC26k Thomee et al. (2016). During testing, we adopt an image-to-GPS retrieval setting: each query image from the test set is matched against a candidate GPS gallery containing 100K (Im2GPS3k) or 500K (GWS15k) coordinates. The evaluation metric is based on the Geodesic Distance, which measures the distance between the predicted and ground-truth coordinates. We report the percentage of predictions that fall within predefined thresholds of 1 km, 25 km, 200 km, 750 km, and 2500 km.

### 5.2 Comparison with State-of-the-art Methods

We conduct a comprehensive comparison between our proposed method and leading global image geo-localization approaches across multiple benchmark datasets, including Im2GPS3k, Google World Streets 15k (GWS15k), and YFCC26k. As shown in Table 1,2,3, our method achieves state-of-the-art performance on Im2GPS3k across all evaluation thresholds (1km, 25km, 200km, 750km, and 2500km), consistently outperforming previous best methods. Notably, our approach demonstrates significant advantages on the more challenging GWS15k dataset, achieving substantial accuracy gains under the 25km, 200km, 750km, and 2500km thresholds compared to prior SOTA models. Compared to GeoCLIP, which does not incorporate textual input, our method shows marked improvement in handling the diversity of global imagery. Even when compared with G3, which also leverages textual information, our approach yields notable performance gains. The GWS15k dataset consists of globally and uniformly sampled image locations, without bias toward any particular region. Moreover, the images exhibit considerable distributional shifts from the training data, making the geo-localization task particularly challenging. Our superior performance on this dataset highlights the effectiveness and strong generalization capabilities of our multimodal alignment framework in complex real-world scenarios.

### 5.3 Ablation Study

To comprehensively evaluate the effectiveness of each proposed component in the tri-modal geo-alignment framework, we conduct a series of ablation experiments. The evaluation metric is Top-1 geolocation accuracy, and all experiments are conducted on the Im2GPS3k dataset Hays & Efros (2008). The results are summarized in Table 4, and we provide detailed analysis in Appendix A.3.

## 6 Conclusion

We present a multimodal framework for global image geolocalization. We first build a tri-modal auto-annotation pipeline—semantic segmentation, vision–language generation, and a Referee module—to produce image-/region-level descriptions with geographic labels. On this data, we propose a unified contrastive learning approach with a shared embedding space, hierarchical alignment, and region-level objectives to jointly model visual, semantic, and spatial cues. To boost accuracy, we design a multi-prompt geographic reasoning module that uses LLMs to generate candidate locations and dynamically expand the GPS gallery. At inference, cross-modal similarity retrieval selects the most likely coordinates. The method achieves SOTA results on multiple benchmarks, validating the effectiveness of multimodal alignment and prompt-driven reasoning for global localization.

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

# A  APPENDIX

## A.1  LLM USAGE STATEMENT

Large language models were an integral component of the proposed framework for global image geo-localization. Specifically, we employed a pretrained vision-language model (OpenAI GPT-4) within our annotation pipeline to automatically generate both image-level and region-level textual descriptions based on semantic segmentation results. These generated captions, together with geographic labels such as city and country names, serve as a textual modality and are embedded into a shared representation space alongside image features and GPS coordinates.

The LLM-generated content is further processed and validated through a referee mechanism to ensure semantic accuracy and consistency before being used in the tri-modal contrastive learning stage. This integration of LLMs is essential to our method's semantic–spatial alignment process and plays a critical role in addressing visual ambiguity and heterogeneous global image distributions. All algorithmic designs, theoretical formulations, training procedures, and experimental analyses were independently developed by the authors. The use of LLMs is fully documented, reproducible, and central to the methodological contributions rather than limited to writing assistance.

### A.1.1  AUTOMATED IMAGE ANNOTATION PIPELINE

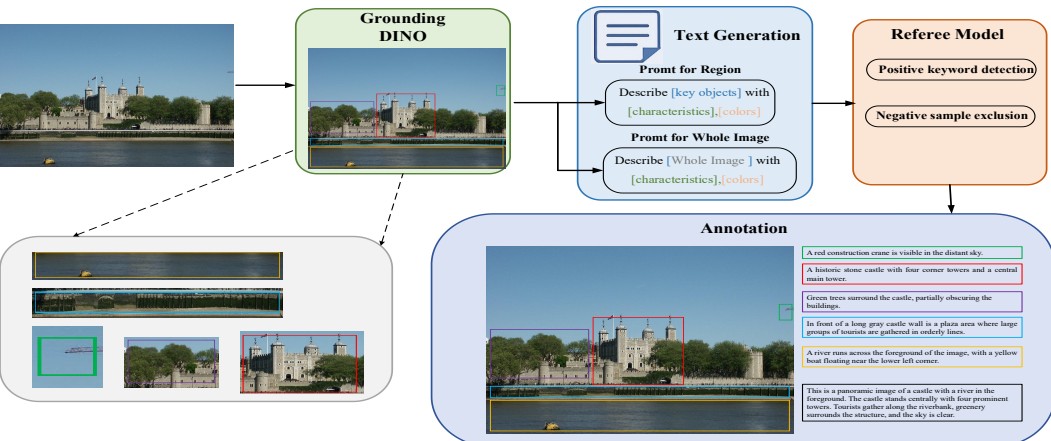

Figure 4: Automated Image Annotation Pipeline. Given an input image, we first apply Grounding DINO to detect multiple semantic regions. For each region and the entire image, a Vision-Language Model generates preliminary region-level and global-level descriptions. These candidate texts are then passed through a Referee Model, which conducts Positive Keyword Detection and Negative Sample Exclusion to filter out vague or hallucinated outputs. The final output consists of structured region-level and global-level descriptions, along with associated geographic labels.

## A.2 EXPERIMENT RESULTS

| Method | Street 1 km | City 25 km % | Region 200 km% | Country 750 km% | Continent 2500 km% |
|---|---|---|---|---|---|
| [L]kNN, $\sigma = 4$Vo et al. (2017) | 7.2 | 19.4 | 26.9 | 38.9 | 55.9 |
| PlaNetWeyand et al. (2016) | 8.5 | 24.8 | 34.3 | 48.4 | 64.6 |
| CPlANetSeo et al. (2018) | 10.2 | 26.5 | 34.6 | 48.6 | 64.6 |
| ISNsMuller-Budack et al. (2018) | 10.5 | 28.0 | 36.6 | 49.7 | 66.0 |
| TranslocatorPramanick et al. (2022) | 11.8 | 31.1 | 46.7 | 58.9 | 80.1 |
| GeoDecoderClark et al. (2023) | 12.8 | 33.5 | 45.9 | 61.0 | 76.1 |
| GeoclipVivanco Cepeda et al. (2023) | 14.11 | 34.47 | 50.65 | 69.67 | 83.82 |
| G3 Jia et al. (2024) | 16.65 | 40.94 | 55.56 | 71.24 | 84.68 |
| Ours | 17.32 | 42.01 | 57.18 | 72.44 | 85.16 |

Table 1: (a) Localization accuracy on the Im2GPS3k Hays & Efros (2008) dataset.

| Method | Street 1 km % | City 25 km % | Region 200 km % | Country 750 km % | Continent 2500 km % |
|---|---|---|---|---|---|
| ISNsMuller-Budack et al. (2018) | 0.05 | 0.6 | 4.2 | 15.5 | 38.5 |
| TranslocatorPramanick et al. (2022) | 0.5 | 1.1 | 8.0 | 25.5 | 48.3 |
| GeoDecoderClark et al. (2023) | 0.7 | 1.5 | 8.7 | 26.9 | 50.5 |
| GeoCLIP Vivanco Cepeda et al. (2023) | 0.6 | 3.1 | 16.9 | 45.7 | 74.1 |
| **Ours** | 0.9 | 4.3 | 18.7 | 49.4 | 78.5 |

Table 2: (b) Results on the GWS15k Clark et al. (2023)dataset.

| Method | Street 1 km % | City 25 km % | Region 200 km % | Country 750 km % | Continent 2500 km % |
|---|---|---|---|---|---|
| PlaNet Weyand et al. (2016) | 4.4 | 11.0 | 16.9 | 28.5 | 47.7 |
| ISNs Muller-Budack et al. (2018) | 5.3 | 12.3 | 19.0 | 31.9 | 50.7 |
| Translocator Pramanick et al. (2022) | 7.2 | 17.8 | 28.0 | 41.3 | 60.6 |
| GeoDecoder Clark et al. (2023) | 10.1 | 23.9 | 34.1 | 49.6 | 69.0 |
| GeoCLIP Vivanco Cepeda et al. (2023) | 11.61 | 22.19 | 36.69 | 57.47 | 76.02 |
| G3Jia et al. (2024) | 23.99 | 35.89 | 46.98 | 64.26 | 78.15 |
| **Ours** | **25.71** | **37.02** | **47.45** | **66.21** | **79.46** |

Table 3: Results on YFCC26k Thomee et al. (2016) dataset

## A.3 ABLATION STUDY

**Location Encoder.** Raw GPS coordinates lie on a non-Euclidean latitude-longitude sphere. Directly using them in a contrastive framework results in spatial discontinuity and unstable gradients. To address this, we introduce a location encoder that transforms coordinates into high-dimensional learnable embeddings. Without this encoder, the model can only align image and text semantically, lacking explicit spatial constraints. As shown in Rows 1–2 of Table 4, the inclusion of the location encoder significantly improves geolocation accuracy. This module provides strong spatial priors and serves as the foundational bridge between semantics and geography, improving modality separation and spatial boundary modeling.

**Textual Module.** Textual descriptions often encode semantic cues that are missing from images or coordinates, offering strong complementary information. However, directly incorporating text may lead to performance drops due to the modality gap. As shown in Rows 2–3 of Table 4, the Top-1 accuracy drops by 3.2% after introducing the textual modality, highlighting the misalignment across modalities. This underscores that while text is semantically rich, a unified representation space is required for effective integration.

**Projection Space.** To mitigate the above modality mismatch, we introduce a learnable nonlinear projection space that aligns image, text, and geolocation embeddings into a unified embedding space. As shown in Rows 3–4 of Table 4, the Top-1 accuracy improves by 7.6%, not only recovering the performance loss from textual inputs, but also surpassing the original bimodal setup. This demonstrates the projection space's capability in harmonizing multimodal distributions.

| Location Encoder | Textual Annotation | Projection Space | Region Contrastive Loss | Global Contrastive Loss | Hierarchical Alignment Loss | Multi-Prompt Geo-Reasoning | Street 1km % | City 25km % | Region 200km % | Country 750km % | Continent 2500km % |
|---|---|---|---|---|---|---|---|---|---|---|---|
| | | | | | | | 10.19 | 26.13 | 34.60 | 48.24 | 63.89 |
| ✓ | | | | | | | 12.78 | 31.45 | 41.20 | 56.70 | 70.20 |
| ✓ | ✓ | | | | | | 10.45 | 28.00 | 36.70 | 50.60 | 66.20 |
| ✓ | ✓ | ✓ | | | | | 13.92 | 36.02 | 47.80 | 63.00 | 78.30 |
| ✓ | ✓ | ✓ | ✓ | | | | 15.01 | 38.91 | 51.23 | 66.42 | 80.76 |
| ✓ | ✓ | ✓ | ✓ | ✓ | | | 15.35 | 39.36 | 54.28 | 68.72 | 82.45 |
| ✓ | ✓ | ✓ | ✓ | ✓ | ✓ | | 16.25 | 40.25 | 54.88 | 69.38 | 83.09 |
| ✓ | ✓ | ✓ | ✓ | ✓ | ✓ | ✓ | 17.32 | 42.01 | 57.18 | 72.44 | 85.16 |

Table 4: Ablation study on the Im2GPS3k dataset Hays & Efros (2008).

**Region Contrastive Loss.**   Global-level contrast often fails to capture fine-grained differences, especially in urban scenes with repetitive structures. We introduce region-level contrastive learning by first segmenting each image into semantic regions and aligning tri-modal embeddings per region. As shown in Rows 4–5 of Table 4, this module significantly improves performance, especially at street- and city-level localization. By mining hard negatives from "intra-image inter-region" and "inter-image geo-neighbors," the model learns to distinguish fine-grained regional differences, proving highly effective in high-redundancy urban contexts. This module enhances local discriminability and pushes beyond the limits of coarse global retrieval.

**Global Contrastive Loss.**   This module performs tri-modal alignment at the global image level, serving as a "coarse-grained semantic anchor" and functioning as the coarse retrieval phase. As shown in Rows 5–6 of Table 4, global contrast improves overall geolocation accuracy, particularly at the country and continent levels. Although region contrast excels at fine-grained discrimination, it lacks macro-level context and may fragment semantic understanding. Removing global contrast loss severely disrupts semantic cohesion, resulting in drifting region embeddings. This module is essential for maintaining global structure and consistent retrieval performance.

**Hierarchical Alignment Loss.**   To ensure consistency between region and image-level representations, we propose a hierarchical alignment loss that minimizes the discrepancy between global embedding and the mean of region embeddings. As shown in Rows 6–7 of Table 4, this module improves performance by 1.6%, and more importantly, stabilizes training. By aggregating region-level features to the global center, it prevents semantic drift and reinforces structural coherence. This is especially useful in scenes with unbalanced region distributions (e.g., open fields vs. commercial blocks), helping weaker regions align better and promoting global-local consistency in representation.

**Multi-Prompt Geo-Reasoning.**   Rows 7–8 in the table demonstrate that the MPGR module brings significantly greater performance gains at higher geographic levels such as Region, Country, and Continent. This indicates that multi-prompt-driven large language models (LLMs) are capable of extracting more abstract and global geographic semantics from image content—such as language cues, architectural styles, and terrain characteristics—which enables more accurate predictions of a given image's broader regional or national origin. This capability is particularly crucial in challenging scenarios where images are ambiguous, cross-domain, or lack distinctive local landmarks. It shows that MPGR is not merely a supplementary signal but a critical component in enhancing the model's global geographic reasoning, providing strong support for hierarchical location inference.

## A.4   PARAMETER ANALYSIS OF MULTI-PROMPT GEO-REASONING

**Analysis of the Impact of Prompt Quantity on Geolocation Performance:** As shown in Fig. 6, increasing the number of prompts from 1 to 7 leads to a general upward trend in geolocation accuracy across all spatial levels, including street, city, region, country, and continent. This indicates that prompt diversification effectively enhances the LLM's ability to reason about the geographic attributes of images. However, beyond four prompts, the improvement in accuracy begins to plateau—especially at finer-grained levels such as street and city. For example, street-level accuracy shows only marginal gains from the 4th to 7th prompt ($17.32 \rightarrow 17.38$), and continent-level accuracy remains nearly unchanged (85.16). It is worth noting that although increasing the number of prompts yields some performance gains, it also introduces considerable computational overhead. More prompts generate more location candidates, which expands the GPS gallery and increases

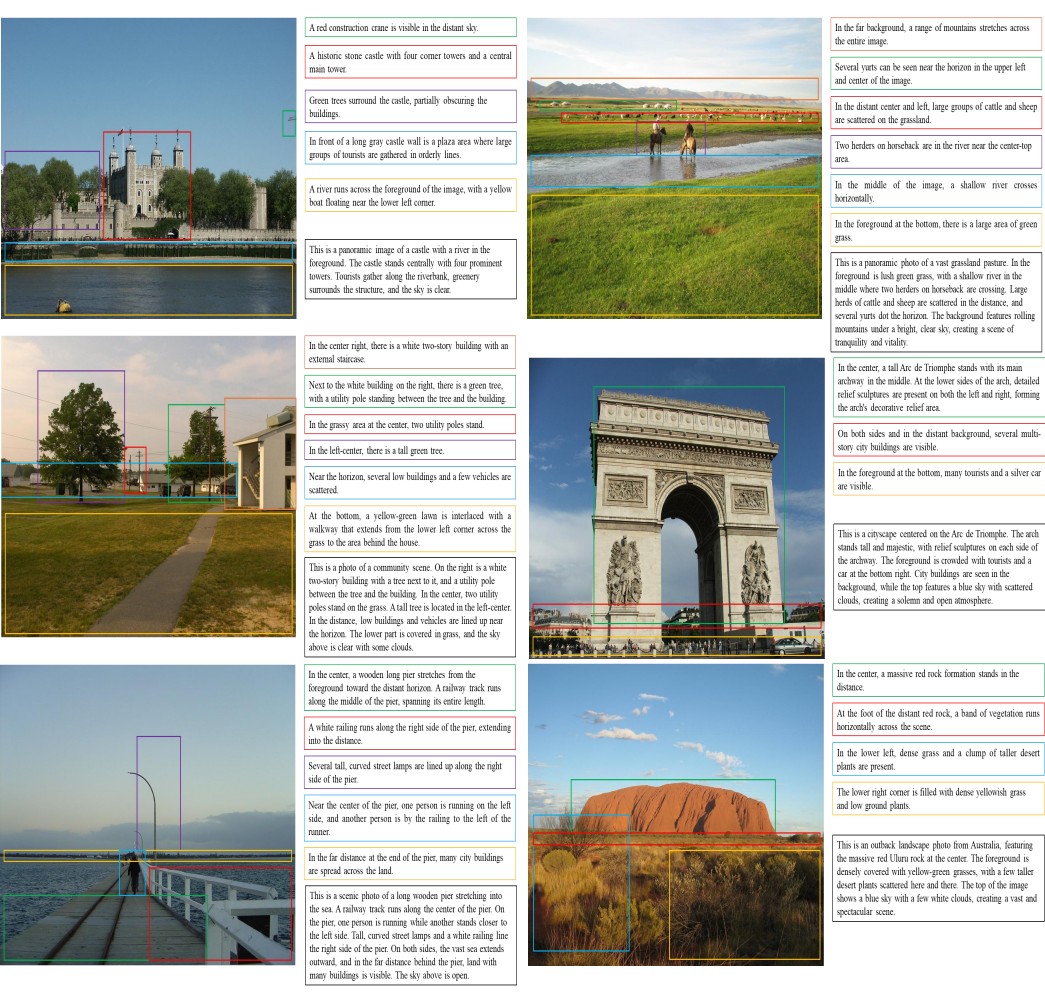

Figure 5: Output Example of the Automated Image-Text Annotation System

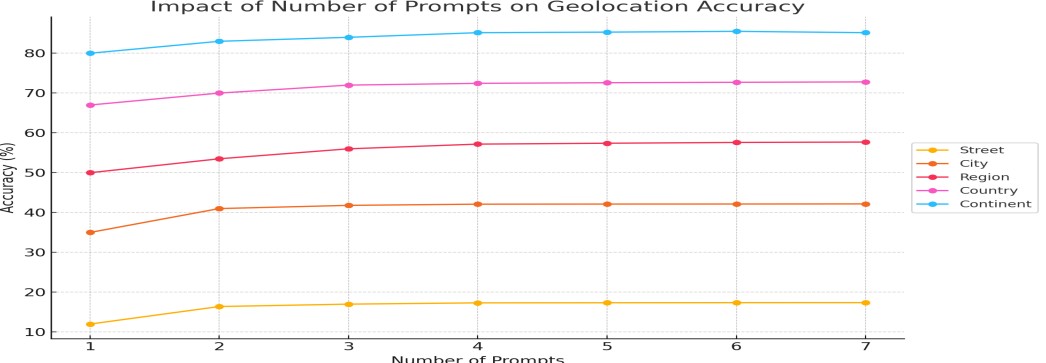

Figure 6: Effect of Prompt Quantity on Geolocation Accuracy.on the Im2GPS3k dataset Hays & Efros (2008)

retrieval time. Additionally, redundant or semantically similar prompts may produce overlapping or ambiguous location predictions, further lengthening the inference pipeline. Therefore, balancing accuracy improvements and computational efficiency, using four prompts offers a practical and effective trade-off.

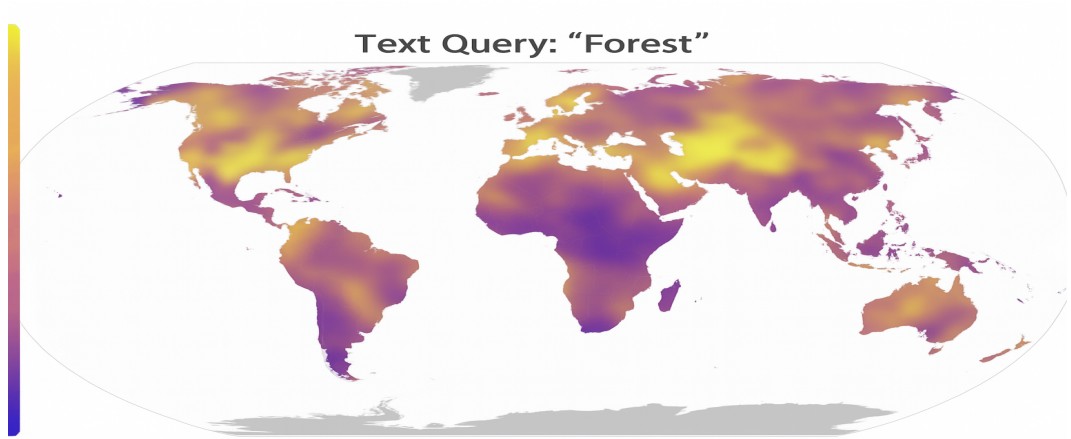

Figure 7: Global geospatial distribution generated from the text query "Forest". The map visualizes cosine similarities between the query's textual embedding and GPS embeddings projected via our Geo-alignment mechanism.

## A.5 GLOBAL GEO-VISUALIZATION FROM TEXTUAL QUERIES

Due to our tri-modal contrastive learning framework, the location encoder is jointly aligned with both visual and textual modalities. This enables us to effectively map free-form textual descriptions into geographic coordinates. This functionality allows the model to assign spatial context to any given textual concept. Specifically, we take an arbitrary text input and extract its embedding using the CLIP text encoder. The embedding is then projected through our trained projection layer and compared against a bank of GPS embeddings. By computing cosine similarity between the text embedding and all GPS coordinate embeddings, we generate a similarity map that reflects the geographic distribution of the queried concept. For example, given the text query "Forest", we visualize the resulting similarity scores over a world map, as shown in Figure 7. The resulting heatmap highlights geographic regions where the learned GPS representations are most semantically aligned with the textual concept of "forest," thereby demonstrating our model's ability to reason about global spatial semantics from purely linguistic inputs.

## A.6 IMPACT OF GEOGRAPHIC LABEL GRANULARITY ON LOCALIZATION PERFORMANCE

| Method | Street 1 km | City 25 km | Region 200 km | Country 750 km | Continent 2500 km |
|---|---|---|---|---|---|
| Continent | 21.99 | 33.89 | 44.98 | 62.26 | 76.55 |
| + Country | 24.02 | 35.60 | 46.80 | 65.31 | 76.93 |
| +City | **25.71** | **37.02** | **47.45** | **66.21** | **79.46** |

Table 5: Localization result under different geographic label granularities on YFCC26k Thomee et al. (2016) dataset

To investigate how different granularities of geographic labels influence image geolocation accuracy, we designed three settings: using only Continent-level labels, using both Country and Continent-level labels, and using full hierarchical labels including City, Country, and Continent. We evaluate model performance on five standard metrics: Street, City, Region, Country, and Continent, with results shown in Table 5. The experimental results demonstrate that increasing label granularity significantly enhances fine-grained localization accuracy. When only Continent labels are used, the model achieves 21.99% on the Street metric. After incorporating City-level labels, performance rises to 25.71%, indicating that fine-grained labels improve the model's ability to perceive local spatial structures. City and Region accuracies also increase by 3.13% and 2.47%, respectively, further confirming the importance of City-level labels in modeling mid- and small-scale geographic semantics. In addition, the use of Country-level labels contributes notably to improvements in Re-

gion and Country metrics, with gains of approximately 1.82% and 3.05%, respectively. Meanwhile, the Continent-level accuracy remains relatively stable, suggesting that coarse labels are sufficient for large-scale region distinction, and finer details offer limited additional benefit at this level. In conclusion, there is a clear positive correlation between label granularity and localization performance. Fine-grained labels are essential for improving performance on high-resolution tasks (e.g., Street and City), while a hierarchical label structure enhances the model's ability to generalize across multiple spatial scales.

## A.7 SATIAL DISTRIBUTION

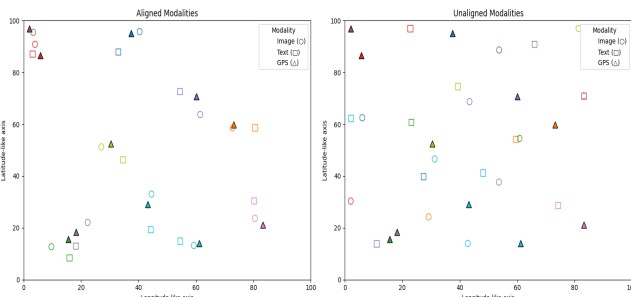

Figure 8: This figure illustrates the spatial distribution of image, text, and GPS modalities under both aligned and unaligned conditions.

Figure 8 illustrates the spatial distributions of image, text, and GPS modalities under both aligned and unaligned conditions. Figure X presents the spatial distribution of the three modalities—image, text, and GPS—in aligned and unaligned states, to intuitively verify the effectiveness of our proposed modality alignment mechanism in geographic semantic learning. We simulate 10 geographic locations (distinguished by color), and represent the three modalities—image, text, and GPS—with different shapes. In both subfigures, we adopt a consistent visualization process: image and text modalities are first passed through their respective feature extractors to obtain embedding features, which are then reduced to two dimensions via t-SNE and linearly scaled to the value range of GPS coordinates to ensure comparability across modalities. The GPS modality is directly visualized using its original coordinates without dimensionality reduction. The left subfigure shows the result after modality alignment. After projection into a shared semantic space, the image and text features are tightly clustered around their corresponding GPS positions, with all three modalities exhibiting clear spatial consistency at the same geographic location. In contrast, the right subfigure shows the result using unprojected features. Here, only the original image and text features are reduced and mapped, and the result shows that although these modalities come from the same geographic locations, their features fail to align with the corresponding GPS points and are instead scattered irregularly across the space, unable to form clusters around GPS anchors. This comparison clearly demonstrates that without feature alignment, different modalities—even when associated with the same location—still exhibit significant spatial deviation, making it difficult to achieve semantic consistency and limiting the feasibility of cross-modal understanding and retrieval. In contrast, with the introduction of a modality alignment mechanism, the three modalities—image, text, and GPS—can establish clear semantic correspondences in a unified embedding space, providing effective support for downstream cross-modal geographic retrieval and location reasoning tasks.

## A.8 ROBUSTNESS ACROSS LANGUAGE MODELS

As shown in Fig. 9, we visualize the predicted geographic coordinates for the same query image using two different large language models: GPT and LLaMA. In the figure, the outer blue circles represent GPT predictions, while the inner red circles represent LLaMA predictions. Despite using different underlying models, the high-confidence predictions from both are geographically close and largely overlapping. Table 6 lists the specific predicted coordinates and their associated confidence scores. Table 7 further compares the performance of our framework when using GPT or LLaMA as the base model across various localization metrics. The results show that although GPT performs slightly better in terms of accuracy, the overall similarity suggests that our framework does not rely on any specific large language model.

| Model | Predicted Location | (Lat, Lon) | Confidence |
|---|---|---|---|
| GPT | Bethany Beach, Delaware | 38.5266, -75.0535 | 22% |
| | Ocean City, Maryland | 38.3750, -75.0700 | 18% |
| | Virginia Beach, Virginia | 36.8500, -75.9780 | 12% |
| | Nags Head, Outer Banks, North Carolina | 35.9096, -75.5970 | 14% |
| | Wrightsville Beach, North Carolina | 34.2592, -77.8286 | 8% |
| | Destin, Florida | 30.3935, -86.4958 | 7% |
| | Gulf Shores, Alabama | 30.2500, -87.6839 | 6% |
| | Cape San Blas, Florida Panhandle | 29.7174, -85.3020 | 5% |
| | Perdido Key, Florida/Alabama border | 30.3250, -87.4367 | 4% |
| | Galveston, Texas | 29.3000, -94.7900 | 4% |
| LLaMA | Bethany Beach, DE | 38.5361, -75.0610 | 20% |
| | Ocean City, MD | 38.4108, -75.0616 | 16% |
| | Cape May, NJ | 38.9351, -74.9083 | 12% |
| | Rehoboth Beach, DE | 38.7109, -75.0890 | 10% |
| | Nags Head (Outer Banks), NC | 35.9408, -75.6724 | 9% |
| | Wrightsville Beach, NC | 34.2158, -77.7976 | 8% |
| | Virginia Beach, VA | 36.8529, -75.9780 | 7% |
| | Gulf Shores, AL | 30.2440, -87.7000 | 6% |
| | Destin, FL | 30.3935, -86.4958 | 6% |
| | Provincetown, MA | 42.0505, -70.1805 | 6% |

Table 6: Comparison of Location Predictions and Confidence Estimates for an Image by GPT and LLaMA.

| Method | Street 1 km % | City 25 km % | Region 200 km % | Country 750 km % | Continent 2500 km % |
|---|---|---|---|---|---|
| LLaMA | 17.23 | 41.94 | 57.07 | 72.35 | 83.82 |
| GPT-mini | 17.15 | 41.67 | 56.89 | 72.21 | 84.68 |
| GPT | 17.32 | 42.01 | 57.18 | 72.44 | 85.16 |

Table 7: Evaluation of localization performance with different LLMs integrated into the Geo-Reasoning pipeline on the Im2GPS3k Hays & Efros (2008) dataset.

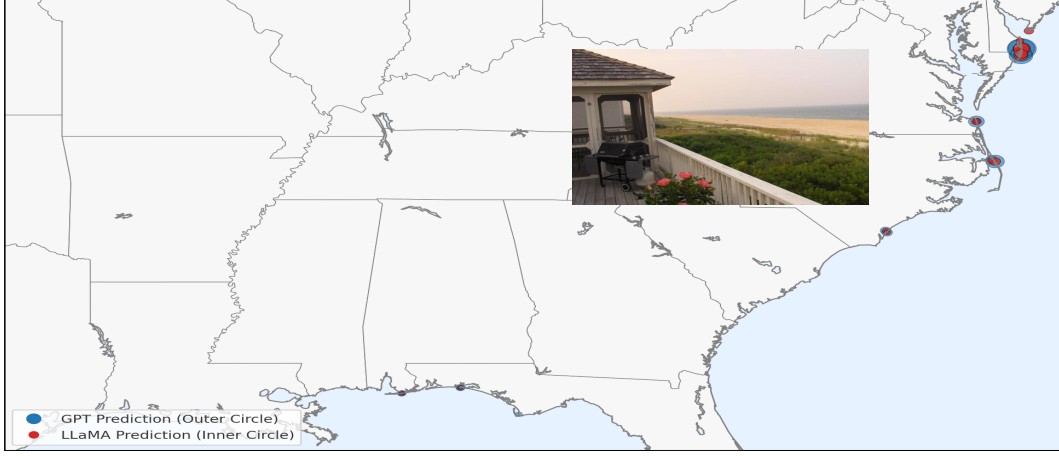

Figure 9: Comparison of location predictions by GPT and LLaMA models. Outer blue circles represent GPT predictions; inner red circles represent LLaMA predictions. Circle size corresponds to model confidence.

