# OpenReview forum: "Beyond GPS: Tri-Modal Contrastive Learning for Global Geo-Localization"
_ICLR.cc/2026/Conference — Submitted to ICLR 2026_

### Official Review · Reviewer_s5iS · 2025-10-22

**Soundness:** 2
**Presentation:** 2
**Contribution:** 2
**Rating:** 2
**Confidence:** 4

**Summary:**

This paper proposes to address the challenging global image geo-localization task by additionally introducing language, which is grounded on the input image. This work consists of a tri-modal annotation pipeline to generate the text for input images and geolocations, and a tr-modal learning pipeline that projects image, text, and GPS into a shared space with contrastive learning. Experiments on three global image geo-localization datasets demonstrate an improved performance compared to previous works.

**Strengths:**

- This paper proposes an interesting method to generate global and regional language descriptions for current image-only geo-localization datasets. The generation pipeline is decomposed into three steps: segmentation, language generation, and validation, whose design is reasonable and convincing.
- Introducing natural language (grounded on the input image) as an additional modality can be an interesting idea to enrich the information utility for image geo-localization.
- Experiments on three datasets demonstrate an improved performance.

**Weaknesses:**

- Using language information to enhance the performance of geo-localization has been explored in previous works. For example, GeoReasoner [1], G3[2]. A discussion of how this works differ from previous studies is missing in the paper.
- In the abstract, it is defined that the image geo-localization task aims to predict the geographic location of a single image, while it is quite confusing to jump into the language modality overlooked by current methods. For the input modality as defined in the task, it has nothing to do with the language modality. In addition, language has already been used in prior geo-localization work, so the claim that current methods “overlook” language seems overstated.
- It is described in section 3.3 that a reverse geocoding module is introduced to encode GPS coordinates as natural language as [continent, country, city]. This raises concerns about loss of positional fidelity: cities span large and irregular areas, so mapping continuous coordinates to coarse categorical labels may degrade location accuracy and introduce boundary artifacts. Also, how this generated text is utilized in the training and inference, how this design affects the performance is not explained clearly in the paper.
- This paper proposes a new location encoder, there are also some other location encoders providing frequency-aware or spherical features for positional embeddings on the Earth surface, for example, the Sphere2vec [3] and spherical harmonics [4]. How the proposed encoder performs against existing encoders has not been investigated or analyzed in the paper.
- Although introducing text information to help with the geolocation is interesting, in what way does the text change the performance can be further explored, for example, does it introduce more geographical clues grounded on the input image? Does it better connect the image modality with the GPS modality as an intermediate modality?
- The implementation details are missing. It is hard to assess if the comparisons with previous works are performed fairly.
- Moreover, a discussion of computational efficiency is also missing in the paper.
-  The paper is poorly structured, and overall it is quite tiring to read the paper and connect the text with figures and tables:
1) Although the authors supplement the results in the appendix while there is no experimental results present in the main paper, making the main paper quite hard to follow.
2) it seems that the line space between sections has been over-tuned since the text looks squeezed together.
 3) images are not correctly ordered, the order of the image jumps from Fig 1 to fig 5 then back to fig 4, and most of the images crucial to explaining the method section are put in the appendix.

Refs:
[1] GeoReasoner: Geo-localization with Reasoning in Street Views using a Large Vision-Language Model. ICLR, 2025.
[2] G3: An Effective and Adaptive Framework for Worldwide Geolocalization Using Large Multi-Modality Models. NeurIPS, 2024.
[3] Sphere2Vec: A general-purpose location representation learning over a spherical surface for large-scale geospatial predictions. ISPRS Journal of Photogrammetry and Remote Sensing, 2023.
[4] Geographic Location Encoding with Spherical Harmonics and Sinusoidal Representation Networks, ICLR, 2024.

**Questions:**

see weaknesses

---

### Official Review · Reviewer_rAa8 · 2025-10-28

**Soundness:** 1
**Presentation:** 1
**Contribution:** 1
**Rating:** 2
**Confidence:** 4

**Summary:**

The paper proposes a pipeline for the task of global image geo-localization.
The pipeline includes a long series of steps, including creating crops based on a semantic segmentation model's output, annotating crops with a VLM to get multiple per-crop descriptions, filtering noisy descriptions using a set of pre-defined positives and negative expressions, using a gallery of embeddings of GPS coordinates, retrieving most similar images, and finally using a location encoder to encode a number of embeddings of GPS coordinates, and comparing those embeddings with embeddings from visual and textual features.

**Strengths:**

The paper obtains state-of-the-art results on multiple popular global localization datasets.

**Weaknesses:**

1. The pipeline is a combination of many models, including an unspecified semantic segmentation model (e.g., SAM + Mask2Former) (line 155), GPT-4, an unspecified vision-geolocation joint encoder (e.g., GeoCLIP) (line 398), and a tuned CLIP model. The GPT outputs captions which often already contain the location of the image (like the Arc de Triomphe and Uluru examples in Figure 5), so it is not clear what is the point of all the other steps in the pipeline, given that GPT-4 can be directly used to estimate a location. Moreover [A] shows that directly using the output of GPT-4.1 achieves 19.1 on Im2GPS3k-1km, while the proposed pipeline achieves 17.32, so it looks like the pipeline is not only much more complex but also produces worse results than directly using GPT-4 output.

2. The comparison with existing models is made even less fair by the fact that the model also uses retrieval over the entire train set.
Comparisons with existing methods are not fair, given that the proposed pipeline uses closed-source VLMs/LLMs (i.e. GPT-4) to predict the location of an image.

3. The paper is hard to read and confusing, due to a large number of modules interacting with each other, and images not being helpful to understand the pipeline. For example, the Multi-Prompt Geo-Reasoning (MPGR) is not clear: according to Figure 1 and Figure 2 the MPGR is a GPT that takes as input the image and captions, and outputs a GPS gallery. I'm assuming the actual output is not the GPS gallery but a series of GPS coordinates, based on Section 4.5. But based on lines 396-400 it looks like the MPGR instead does image retrieval on the training set using a model like GeoCLIP (line 398).

4. Figure 2 does not help to understand the GPS Gallery Construction from Multi-Prompt Outputs: there seem to be random lines without directions between different entities (two separate lines between the grey GPS and GPT), the line between the green GPS intersecting the one between the image and the GPT for unclear reasons. And line 393 refers to Figure 3, although I believe it should refer to Figure 2, because Figure 3 does not seem to be representing the framework of MPGR.

5. No result is presented in the entire paper, forcing the reader to read the Appendix.

[A] Grainge et al, Assessing the Geolocation Capabilities, Limitations and Societal Risks of Generative Vision-Language Models

**Questions:**

Authors can address the weaknesses mentioned above. Addressing some of them (like readability and figure clarity) could help improve the paper, although I don't think there is a way to make the results comparable with previous methods: if anything, the paper should compare with other pipelines using closed-source LLMs, given that the proposed pipeline uses GPT-4 to provide clues about the location, or directly the location itself.

It would also be good to report inference time of the entire system, which seems like it could be considerably high given the many modules involved.

The abstract says that global image geo-localization is applicable in robotic navigation, although the task is not mentioned throughout the entire paper. It seems counterintuitive that a robot would need global image geo-localization, and the paper does not offer examples of this happening. I would suggest to either discuss this more, provide some sources, or remove it from the abstract.

---

### Official Review · Reviewer_h9kv · 2025-10-29

**Soundness:** 2
**Presentation:** 2
**Contribution:** 2
**Rating:** 2
**Confidence:** 4

**Summary:**

This paper proposes a three-way modality contrastive learning framework for global image geolocalization that unifies image, text, and GPS modalities within a shared embedding space. The authors build an automated annotation pipeline combining semantic segmentation (both global and local), vision-language generation, and a referee mechanism to produce high-quality region- and image-level textual descriptions linked with GPS coordinates. A hierarchical contrastive learning objective aligns semantics at both global and regional scales, while a multi-prompt LLM reasoning module expands the GPS candidate gallery for robust inference.

**Strengths:**

The core motivation of this paper lies in introducing rich semantic information for images, which enhances the representation quality in global geolocalization. In addition, the paper proposes several innovations in GPS modeling and optimization strategies. The step of incorporating rich semantic cues into geolocation is particularly interesting and adds meaningful depth to the framework.

**Weaknesses:**

1. The related work section misses many recent studies, and the Location Prediction from Images subsection does not include papers from 2024 or 2025.
2. In line 176, what is the accuracy of the binary classification model? Adding more experiments about this will be better.
3. In line 208, references should be added for InfoNCE and triplet loss.
4. In line 211, why is EEP used? Is there any experimental evidence showing that EEP is necessary?
5. In line 261, which prior works are being referred to? The citations are missing.
6. In line 327, is there any experiment proving that pairwise alignment among the three modalities performs better than aligning two modalities to one reference (e.g., only (I,T) and (I,G))?
7. In Equation (9), the hyperparameter experiments for the three λ coefficients are missing.
8. In Section 4.5, which large model is used during inference?
9. In line 399, credit should be given to previous work such as Img2Loc, since the approach is similar.
10. In Section 4.6, is there any experiment demonstrating that this method outperforms random sampling?
11. In Section 5, the experimental section is too brief, and most results are only in the appendix, which makes the paper difficult to read. Additionally, key experiments such as hyperparameter analysis are missing. Although some extra experiments appear in the appendix, sensitivity studies on λ-weighting for the three losses are not included.
12. Regarding GWS15k, since this dataset is not publicly available, how was it constructed, and will it be released for reproducibility?
13. It would be helpful to add more implementation details, such as GPU type, training time, and other experimental settings, to facilitate future research.

**Questions:**

Please refer to the weakness section

---

### Official Review · Reviewer_9NcX · 2025-10-30

**Soundness:** 3
**Presentation:** 3
**Contribution:** 3
**Rating:** 6
**Confidence:** 3

**Summary:**

This paper proposes a novel tri-modal contrastive learning framework integrating image, text, and GPS data for worldwide image geo-localization. The authors address the limitations of existing methods that rely solely on visual or dual-modal inputs, by introducing geographic semantics through language. The proposed system includes an automated annotation pipeline combining semantic segmentation, vision-language generation, and a referee model for quality control. A unified projection space aligns the three modalities, trained with hierarchical global–regional contrastive objectives and dynamic hard negative mining. Furthermore, a Multi-Prompt Geo-Reasoning (MPGR) module utilizes large language models (LLMs) to generate diverse geographic hypotheses and dynamically expand a GPS gallery during inference. Extensive experiments on multiple benchmarks (Im2GPS3k, GWS15k, YFCC26k) demonstrate state-of-the-art performance and robust generalization.

**Strengths:**

1. Tri-modal framework (Image–Text–GPS): The paper pioneers a unified tri-modal contrastive learning framework for geo-localization, effectively bridging visual, textual, and geographic spaces.

2. Automated annotation pipeline: Introduces a high-quality image–text–GPS dataset construction process combining semantic segmentation, vision-language models, and a “Referee Model” for filtering hallucinated or low-quality captions.

3. Unified projection space with hierarchical contrastive objectives: Establishes a shared embedding space where all modalities are jointly aligned using global–region contrastive learning, hierarchical consistency loss, and dynamic hard negative mining.

4. Multi-Prompt Geo-Reasoning (MPGR): Proposes a novel prompt-driven reasoning mechanism leveraging LLMs to generate geographically diverse candidate coordinates, improving inference robustness and reasoning interpretability.

**Weaknesses:**

1. Dependence on large language models – The annotation and MPGR modules heavily rely on LLMs (e.g., GPT-4), which may introduce bias, reproducibility issues, or cost barriers for deployment.
2. Limited efficiency analysis – The paper lacks a detailed analysis of computational cost, especially regarding the MPGR’s multi-prompt inference and large GPS gallery construction.
3. Insufficient ablation on text modality contribution – While textual information is central to the framework, the specific quantitative impact of text versus image–GPS alignment could be further clarified.
5. Missing Related Works: The review of related literature is not comprehensive. Several recent works on text–image joint geo-localization and multimodal matching are not cited or discussed, such as:
[1] Lu X, Zheng Z, Wan Y, et al. GLEAM: Learning to Match and Explain in Cross-View Geo-Localization, arXiv:2509.07450, 2025.
[2] Ye J, Lin H, Ou L, et al. Where am I? Cross-View Geo-localization with Natural Language Descriptions, arXiv:2412.17007, 2024.
[3] Sun J, Huang J, Jiang X, et al. CGSI: Context-Guided and UAV’s Status Informed Multimodal Framework for Generalizable Cross-View Geo-Localization, IEEE TCSVT, 2025.

**Questions:**

None

---

### Meta-Review · Area_Chair_Bqmz · 2026-01-08

**Summary:**

This paper proposes a tri-modal (image–text–GPS) contrastive learning framework for global image geo-localization, augmented with an automated annotation pipeline and an LLM-based Multi-Prompt Geo-Reasoning (MPGR) module. Some reviewers appreciate the ambition of integrating language semantics into geo-localization and note strong benchmark performance. However, the reviews raise serious concerns regarding methodological clarity, experimental fairness, and scientific validity.

A central issue is the heavy reliance on closed-source LLMs (e.g., GPT-4) across multiple stages, including annotation, reasoning, and inference, making it difficult to disentangle learned representation quality from direct LLM prior knowledge. Several reviewers also find the pipeline overly complex, poorly specified, and insufficiently justified, with unclear ablations isolating the contribution of each component. Concerns about fairness of comparisons, missing efficiency analysis, incomplete related work coverage, and limited reproducibility further undermine confidence in the results.

**Reviewer Concerns:**

Most major concerns remain unresolved after discussion. Reviewers consistently questioned whether the reported performance gains stem from genuine tri-modal representation learning or from direct leakage of location information via LLM-generated captions and prompts. The use of GPT-4 both for annotation and inference raises fairness and comparability issues, especially when competing methods do not have access to such signals.

Additional unresolved issues include lack of clear ablation studies, insufficient analysis of computational cost (particularly for MPGR), missing or incomplete citations to recent related work, and unclear dataset construction details (e.g., for non-public benchmarks). While the authors clarify some implementation details, these responses do not sufficiently address the core concern that the pipeline’s effectiveness is inseparable from proprietary LLM capabilities, limiting scientific insight and reproducibility.

**Reviewer Scores:**

Reviewer 9NcX (initial: 6): Likely remains 6, positive but cautious.

Reviewer h9kv (initial: 2): Likely remains 2, with many technical concerns unresolved.

Reviewer rAa8 (initial: 2): Likely remains 1–2, given strong objections to fairness and clarity.

---

### Decision · Program_Chairs · 2026-01-26

Reject